# Identification of the *BZR* Family in Garlic (*Allium sativum* L.) and Verification of the *AsBZR11* under Salt Stress

**DOI:** 10.3390/plants13192749

**Published:** 2024-09-30

**Authors:** Xianghan Peng, Jiaojiao Ruan, Fangling Jiang, Rong Zhou, Zhen Wu

**Affiliations:** 1College of Horticulture, Nanjing Agricultural University, Nanjing 210095, China; 2021104071@stu.njau.edu.cn (X.P.); 2023204045@stu.njau.edu.cn (J.R.); jfl@njau.edu.cn (F.J.); zhour@njau.edu.cn (R.Z.); 2Department of Food Science, Aarhus University, Agro Food Park 48, 8200 Aarhus, Denmark

**Keywords:** garlic, *AsBZR* gene family, expression profile, salt stress, functional validation

## Abstract

Brassinazole-Resistant (BZR) is an important transcription factor (TF) in the brassinosteroid (BR) signaling pathway, which plays a crucial role in plant growth, development and stress resistance. In this study, we performed a genome-wide analysis of *BZRs* in garlic (*Allium sativum* L.) and identified a total of 11 members of the *AsBZR* gene family. By comparing the expression patterns of *AsBZR* genes under salt stress, the candidate gene *AsBZR11* with salt tolerance function was identified. Subcellular localization results showed that *AsBZR11* was localized in the nucleus. The salt tolerance of overexpression lines improved, and the germination rate and root length of overexpression lines increased as compared with wild type. The content of reactive oxygen species (ROS) decreased, and the activity of antioxidant enzymes increased in *AsBZR11*-OE, suggesting that *AsBZR11* has the function of improving plant salt tolerance. Our results enriched the knowledge of plant *BZR* family and laid a foundation for the molecular mechanism of salt tolerance of garlic, which will provide a theoretical basis for the subsequent creation of salt-tolerant germplasm resources.

## 1. Introduction

Garlic (*Allium sativum* L.) is annual and biennial herb of the *Allium* in the *Liliaceae*, which can be both edible and medicinal [1]. Salt stress is an unfavorable environmental factor that severely limits agricultural production [2]. Garlic is a shallow-rooted, moisture-loving vegetable and is highly susceptible to abiotic stresses including salt, which adversely affects yield and quality of garlic [3].

Brassinosteroids (BRs), as phytohormones, play critical roles in regulating plant response to abiotic stress, seed germination, photomorphogenesis and cell elongation, etc. The BR is a growth-promoting steroid hormone, which belongs to the six plant hormones together with growth hormone (IAA), cytokinin (CTK), ethylene (ETA), gibberellin (GA) and abscisic acid (ABA) [4]. BR can alleviate the damage of abiotic stresses on plants [5]. For example, the exogenous application of BR was able to reduce the negative effects of drought stress on apple (*Malus domestica* L.) [6]. The BR can alleviate lipid membrane peroxidation in mint (*Mentha piperita* L.) under salt stress, reduce the adverse effects on growth and development and increase the essential oil content of mint under salt stress [7]. Meanwhile, BR plays an important role in various physiological processes of development and metabolic synthesis in the plant, such as cell elongation, flowering, senescence and leaf morphogenesis [8].

Brassinazole-resistant transcription factor (*BZR*) is a family of transcription factors being located downstream of the BR signaling pathway, which regulates plant growth and development at abiotic stress by altering the expression of BR-responsive genes. In *Arabidopsis*, many *BZR* genes have been reported, and *BZR1* can control the expression of BR-regulated genes. *BZR1* is also a major transcription factor in the regulation of growth and development in plants, with a similar function of *BZR2* [9,10]. The *BZR* family has already been identified in other plants, such as tomato, rice, etc. [11,12,13]. The rice BZR1/BES1 gene family in *Oryza sativa* L. plays a role in BR and ABA signaling pathways and responds to salt stress.

Garlic, an important economic crop, is widely cultivated worldwide. China is the world’s major garlic producer, consumer and exporter. Among *Allium*, garlic is the first species to complete whole genome sequencing and assembly, which provides an important reference for molecular genetic research of *Allium* plants. The public availability of garlic genomic data makes it possible to identify the *AsBZR* family. As an important transcription factor, BZR is a key regulator of the BR pathway. However, few studies related to the garlic *BZR* gene family have been reported. The *BZR* family is functionally diverse, and thereby it is important to study *AsBZR*. Herein, we performed bioinformatic analysis of the *BZR* family gene members in garlic. The expression patterns of *AsBZRs* in different organs under salt stress treatments were analyzed, and *AsBZR11* was identified as the most significant gene in response to salt stress. Moreover, we investigated the *AsBZR11* protein through subcellular localization. The biological functions of *AsBZR11* were clarified by characterizing the physiological responses of the transgenic lines under salt stress, where Agrobacterium was applied to obtain *Arabidopsis* overexpression lines transduced with *AsBZR11*. This study will explore the function of the *BZR* gene family in improving plant salt tolerance and lay the foundation for further research on the molecular mechanism of *AsBZR* in salt tolerance improvement of garlic.

## 2. Results

### 2.1. AsBZR Family Members and Nomenclature

A total of 11 *BZR* members were identified in garlic (Figure 1). They were named as *AsBZR1*~*AsBZR11* according to the previous nomenclature guideline [14]. *AsBZR1* was found to be located on top of chromosome 1, whereas *AsBZR2*, *AsBZR3* and *AsBZR4* were distributed in chromosome 2, and *AsBZR5*, *AsBZR6* and *AsBZR7* were found on chromosome 6. In addition, *AsBZR8*, *AsBZR9* and *AsBZR10* were localized on chromosome 7, and *AsBZR11* was located on scaffold17110.

### 2.2. Evolutionary Relationships of the AsBZR Gene Family Members

In order to study the evolutionary relationship between *BZR* genes from different plant species, MEGA was used to construct *BZR* family evolutionary trees for a total of four species, including garlic, *Arabidopsis thaliana*, tomato and rice. As shown in Figure 2, *BZR* were divided into five subfamilies, where *AsBZR* genes were distributed in all five groups, showing different affinities with the four species.

### 2.3. Characterization of BZR Expression in Garlic

#### 2.3.1. Expression Pattern of *AsBZR* Genes under Salt Stress

*AsBZR5* was induced by the salt stress in all measured time points, and the rest of the genes were up-regulated part of the time (Figure 3). The relative expression of *AsBZR1* and *AsBZR2* peaked at the 24 h, and that of *AsBZR9* was the highest at 12 h. There were two peaks in the expression of *AsBZR4* and *AsBZR10*, and the expression of *AsBZR4* was significantly higher at 6 h and 24 h of salt treatment than at other times. *AsBZR3*, *AsBZR6*, *AsBZR7*, *AsBZR8*, *AsBZR9* and *AsBZR11* all had one peak of expression, and the peaks of *AsBZR3*, *AsBZR6*, and *AsBZR7* appeared at 24 h. Moreover, the *AsBZR9* and *AsBZR11* peaks appeared at 12 h and 72 h, respectively. Significant changes in the relative expression of *AsBZR* genes indicated that they might be involved in the salt tolerance response of garlic, among which *AsBZR11* expression was up-regulated by nearly 40-fold at 72 h after salt treatment than control. It was hypothesized that *AsBZR11* might play an important role in the salt tolerance of garlic, which needs to be further verified.

#### 2.3.2. Tissue-Specific Expression Patterns of *AsBZR* Genes

Gene function was closely related to its expression pattern. The tissue-specific expression of 11 *AsBZR* genes in different organs was investigated using qRT-PCR (Figure 4). *AsBZR* genes were expressed in the roots, pseudostems, leaves and bulbs of garlic. Compared with other members, the transcription levels of *AsBZR1*/*3/4* were highly expressed in roots. *AsBZR2* and *AsBZR9* highly expressed in the pseudostem. *AsBZR6* was the highest in the leaves, and *AsBZR5/7/8/10/11* genes were highly expressed in the bulbs of garlic. The relative expression of *AsBZR11* was significantly higher than that of other tissues. *AsBZR* genes plays an important role in the development process of garlic tissues.

### 2.4. AsBZR11 Protein Was Located in the Nucleus

To address the possible functions of AsBZR11, subcellular localization was performed. The vector bearing the fusion construct PRI101-AsBZR11–GFP and the control GFP vector were individually transformed into leaves of Nicotiana benthamiana using a 2 mL needleless syringe. Green fluorescence was observed in all tobacco cells after injection of the PR101-GFP strain into tobacco, which appeared in the nucleus site after injection of AsBZR11-PR101-GFP (Figure 5). Confocal laser scanning microscopy analysis showed that the AsBZR11 protein was mainly located in the nucleus region.

### 2.5. Identification of AsBZR11-OE Lines

In order to reveal the function of *AsBZR11* in garlic responding to salt stress, we constructed an overexpression vector for transgene. After infection, the *Arabidopsis* seeds were sterilized and flattened in kanamycin-containing medium, vernalized and then cultivated normally. Positive seedlings with normal growth were screened out at about 14 d, transferred to the substrate for further cultivation, and the DNA of the plants was extracted and detected by PCR assay. The results of the DNA assay and the relative expression are shown in Figure 6. Further RNA extraction and RT-qPCR identification showed that the expression level of *AsBZR11* in overexpression plants was higher than that in WT, indicating that *AsBZR11* gene was successfully transferred into *Arabidopsis thaliana*. Highly expressed lines were selected and cultured as homozygotes for further study.

### 2.6. Seed Germination and Seedling Phenotyping of AsBZR11 Transgenic Arabidopsis under Salt Stress

We tested the salt tolerance of transgenic lines. There was no significant difference in root length and germination between wild-type and transgenic lines on normal MS medium (Figure 7). However, the root length of transgenic lines was significantly higher than that of the wild type at NaCl treatment. In addition, the germination rate of overexpression plants on salt-stressed medium was significantly higher than the wild type. Under salt stress, the development of the transgenic lines was better than that of the wild type, indicating that the transgenic lines had stronger salt tolerance.

In order to further verify the function of the transgenic lines, the *Arabidopsis* seedlings were treated by salt stress. After salt stress was applied to transgenic *Arabidopsis* seedlings for 10 days, both *AsBZR11* overexpression and WT plants can not stay green with wilted leaves, where more severe damage were observed on WT plants (Figure 8). This indicated that the transfection of *AsBZR11* alleviated the salt stress injury and enhanced the salt tolerance.

The differences in superoxide anion (O_2_^•−^) generation rate and hydrogen peroxide (H_2_O_2_) content in wild-type and transgenic *Arabidopsis* were not significant (Figure 9). Under salt treatment, the ROS content of overexpression lines was significantly reduced compared with the wild type, suggesting that the *AsBZR11* transgenic plants were tolerant to salt.

In order to alleviate the oxidative damage caused by stress, plants regulate the activity of antioxidant enzymes in cells via metabolic activities. There was no significant difference in antioxidant enzyme activities between *AsBZR11* overexpression lines and WT under control conditions (Figure 10). After salt stress, the activity levels of Superoxide dismutase (SOD), Catalase (CAT), Peroxidase (POD) and Ascorbate peroxidase (APX) in transgenic plants were significantly higher than those in WT under salt treatment. This indicated that the overexpression of *AsBZR11* in *Arabidopsis thaliana* could effectively enhance the activity levels of its antioxidant enzymes, thus reducing the damage caused by reactive oxygen species to the plants, and further improving the ability of plants to tolerate salt stress.

There were no significant differences in relative conductance, MDA content and proline content between control transgenic plants and wild-type plants (Figure 11). However, after salt treatment, the Malondialdehyde (MDA) content of wild-type *Arabidopsis* was significantly higher than those of the overexpression lines, demonstrating that *AsBZR11* overexpression plants had lower membrane permeability than WT and was less damaged under stress. The REC was significantly lower than WT, and overexpression of *AsBZR11* could reduce the damage of cell membrane caused by salt stress, reduce the degree of membrane lipid peroxidation and improve the salt tolerance of garlic. The proline content in *AsBZR11* overexpression plants was significantly higher than that in the wild type under salt stress conditions. The proline content could represent the plant’s stress tolerance to a certain extent, suggesting that the overexpression lines had a stronger ability to withstand salt stress injury.

## 3. Discussion

The *BZR* transcription factor is a positive regulator of BR signaling. The *BZR* family is essential for plant growth and development, which is also involved in the plant stress response [12,15]. The *BZR* family has been reported in a variety of plants. For example, rice *OsBZR1* is involved in the response of rice to BR and thereby regulates rice growth and development [13]. A total of 52 *BZR* genes were identified in seven legumes, many of which regulate organ differentiation and respond to abiotic stresses such as drought and salt [16]. *TaBZR2* mediates the interaction between BR and drought signaling pathway [17].

*BZR* genes were differentially expressed in different parts of the plant and at different stages of growth and development. The abundance of *BZR* gene transcripts significantly increased at the tasseling stage in rice, and all members except *OsBZR1* had high expression levels in the healing tissues, pistils and roots [18]. In wheat, all *TaBZRs* were highly expressed in stems and spikelets, and some family members also had high expression levels in internodal tissues [16]. This suggested that *BZR* genes play a role in regulating a variety of growth and development in rice and wheat [18]. Many *BZR* genes regulate organ development and differentiation in legumes and respond significantly to drought and salt stress [16]. *SlBZR* genes significantly differentially expressed in various tissues and organs at different stages of growth and development in tomato, and some of the genes had tissue-specific expression [19]. In this study, 11 members of the garlic BZR family were identified, and the bioinformatics characteristics such as protein physicochemical properties, conserved domains and cis-acting elements in the promoter region were analyzed. The *AsBZRs* were differentially expressed in various organs of garlic and may play different roles in regulating the growth and development of each organ. Under salt stress, *AsBZR11* and *AsBZR8* were screened as candidate genes, since *AsBZR11* and *AsBZR8* could improve the salt tolerance of plants.

Genes responsible for regulating protein synthesis may undergo changes in expression due to external stimuli when the environment changes [20]. *BZRs*, as key genes regulating protein synthesis, undergo changes in expression when the growth environment changes, e.g., *NnBZR1.2* in Lotus responded significantly to abiotic stresses such as low temperature, drought and the induction of Cd and NaCl [21]. In this study, the expression of *AsBZRs* also changed to different degrees after salt stress was applied to garlic plants, of which *AsBZR11* up-regulated the most significantly.

Genetic transformation of plants can change the genetic material of plants at the genetic level, thereby improving plant traits. By introducing the *SlBZR1D* gene from tomato, *Arabidopsis* plants were more sensitive to the growth regulator BR, and their tolerance to salt stress was also enhanced [22]. After overexpression of *SlBZR1D*, the MDA content in *Arabidopsis* significantly reduced, and the chlorophyll content significantly increased [22]. This indicated that the overexpression lines may have less cell membrane damage and higher POD and CAT activity than the wild type [23]. These results were consistent with our findings.

The result show that *35s: AsBZR11* showed improved salt tolerance, the germination rate and root length of *AsBZR11* overexpression lines significantly increased under salt stress. One of the major intrinsic manifestations of plant response to abiotic stress is ROS accumulation, and ROS is also a major regulator of plant response to a variety of abiotic stresses [24,25]. The transfer of *AsBZR11* significantly reduced the accumulation of ROS in *Arabidopsis* plants under salt stress and increased the activity of antioxidant enzymes, which could enhance plant tolerance [26,27]. MDA characterizes the degree of lipid peroxidation in the cell membrane [28]. Relative conductivity reflects the degree of damage to plant cells [29]. Here, the *AsBZR11* transgenic plants showed lower MDA levels and relative conductivity under salt stress, which indicated that their cell membrane permeability was weakened and plant tolerance to stress was improved. Proline is an important intracellular osmoregulatory substance, and its content can be used as an indicator of plant stress resistance [30]. The proline content of *AsBZR11* transgenic *Arabidopsis* was significantly higher than wild type under salt stress, suggesting that the transfection of *AsBZR11* improved the accumulation of osmotic substances at stress condition. Compared with the wild type, the transgenic lines had less ROS production, lower MDA content and higher proline content in *Arabidopsis*. The overexpression lines increased the transcript level of *AsBZR11*, resulting in less ROS production, lower MDA content and higher proline content, thereby improving the salt tolerance of the plants.

## 4. Materials and Methods

### 4.1. Identification and Cloning of AsBZR Genes

Sun et al. (2020) sequenced and assembled the garlic genome to obtain a highly complete garlic genome map, and 16.9 GB of garlic genome was detected [31]. To identify a complete list of garlic RBZ genes, the nucleotide and amino acid sequences of the predicted *AsRBZ* genes were downloaded via https://doi.org/10.1016/j.molp.2020.07.019 [31]. The HMM profile of the RBZ conserved domain (PF05687) was downloaded from the PFAM database (http://pfam.xfam.org) and used to survey all proteins on 25 June 2023. The ExPasy website (http://web.expasy.org/protparam/) was used to survey protein length, molecular weights and isoelectric points of deduced polypeptides on 28 June 2023. The *BZR* gene family database was downloaded from the Database of *Arabidopsis* (http://datf.cbi.pku.edu.cn/), rice (https://www.ricedata.cn/gene/) and tomato (https://solgenomics.net/) on 11 July 2023.

### 4.2. Plant Growth Conditions and Salt Stress Treatment

*Allium sativum* L. ‘Jinxiang’ garlic were used as plant materials. Garlic bulbs were grown in a climate chamber (14/10 h light/dark regimen, 25 °C/18 °C, photosynthetically active radiation of 360 µmol m^−2^ s^−1^ and 70% relative humidity).

The salt stress treatment was performed using the seedling with five leaves. The seedings were irrigated using 2 L nutrient solution with 200 mmol·L^−1^ NaCl and 0 mmol·L^−1^ NaCl as salt treatment and control, respectively. Samples were collected after 0, 3, 6, 9, 12, 24, 48 and 72 h of treatment for real-time fluorescence quantitative PCR (qRT-PCR) assay. Three biological replicates were taken at each time point and three mixed samples were taken for each replicate.

### 4.3. Garlic RNA Extraction and qRT-PCR

Total RNA was extracted using Trizol (Invitrogen, Waltham, MA, USA) and reverse transcribed to cDNA by HiScript IV 1st Strand cDNA Synthesis Kit (Novizen, Nanjing, China). The primers were designed by Primer 5.0 (Table 1). qRT-PCR was conducted using an Quantstudio 3 real-time quantitative fluorescent PCR apparatus (AppliedBiosystems, Foster City, CA, USA) with the TOROIVD qRT Master Mix kit (Toroivd, Shanghai, China). The procedure was performed as follows: 95 °C 60 s; 95 °C 10 s, 60 °C 30 s, 40 cycles. The relative expression levels were calculated using the 2^−∆∆Ct^ method with *AsACTIN* as a reference gene.

### 4.4. Subcellular Localization of AsBZR11 Protein

The primer sequences for subcellular localization are shown in Table 2. The coding sequence of *AsBZR11* without the stop codon was cloned into PR101 to generate the 35S::AsBZR11-GFP fusion gene. The 35S::AsBZR11-GFP and control GFP were individually transferred into leaves of *Nicotiana benthamiana*. The plants incubated in the dark for 12 h. The suspension contained 10 mM MgCl_2_, 150 μM As and 10 Mm MES (PH 5.7), and the OD600 was adjusted to about 0.8. Images of transformed tobacco cells were captured using a ×40 objective lens with bright field and GFP.

### 4.5. Overexpression Vector Construction and Plant Transformation

The primer sequences for overexpression vector are shown in Table 3. The coding sequence of *AsBZR11* was cloned into the pBI121 vector, and the *AsBZR11*-pBI121 recombinant plasmid was transformed to GV3101.

Wild *Arabidopsis* seeds were sterilized, and the seeds were spread flat on MS plates, which were firstly subjected to low temperature vernalization followed by normal incubation, and then transplanted into sterilized substrates 2 weeks later. *AsBZR11* was transfected into *Arabidopsis* using the flower-dipping method during the flowering period. After harvesting the *Arabidopsis* seeds, the positive seedlings were screened using screening medium, and the DNA was extracted using CTAB method for positive identification. The positive lines were cultured to T_3_ generation for purification for all experiments.

### 4.6. Transgenic Plants Salt Stress Treatment and Physiological Indexes Determination

Overexpressed lines and wild-type *Arabidopsis* were vernalized in MS medium containing 200 mg·L^−1^ NaCl and 0 mg·L^−1^ NaCl, respectively. The root length of the *Arabidopsis* seedlings was measured, and the germination rate was counted after 1 month. The samples were taken for the subsequent determination of physiological indexes.

The O_2_^−^ generation rate and H_2_O_2_ content was determined according to the method from Ke et al. (2007) and potassium iodide spectrophotometry [32,33], respectively. The SOD, POD, CAT and APX activities were determined using nitrogen blue tetrazolium (NBT) method, the guaiacol method, the methods from Aebi (1984) and the method of Nakano et al. (1981), respectively [34,35,36,37]. The REC, MDA and proline content were determined according to the method of Kong et al. (2008), thiobarbituric acid method and ninhydrin colorimetric method, respectively [38,39,40].

### 4.7. Data Analysis

The data were analyzed using Excel 2007 and IBM SPSS Statistics 25.0 software. The significance of the differences between the different time points was tested using Tukey HSD, *p* < 0.05, and graphs were plotted using GraphPad prism 8.0.

## Figures and Tables

**Figure 1 plants-13-02749-f001:**
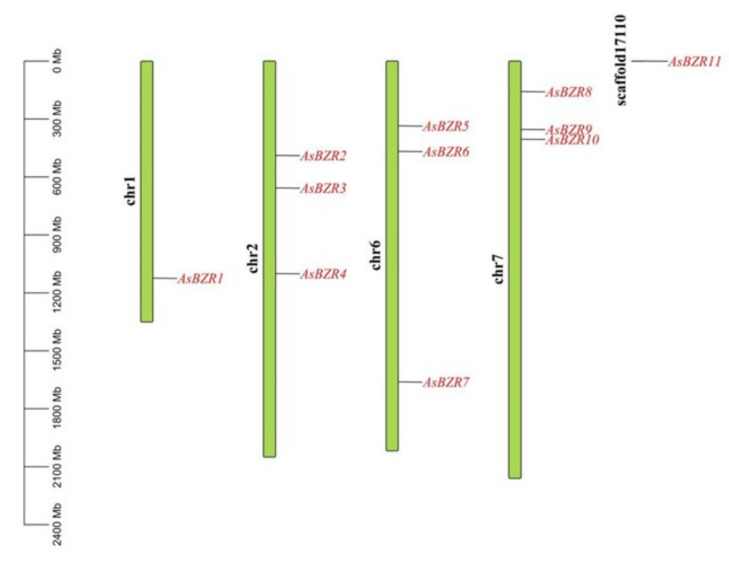
Chromosomal location of *AsBZR* genes.

**Figure 2 plants-13-02749-f002:**
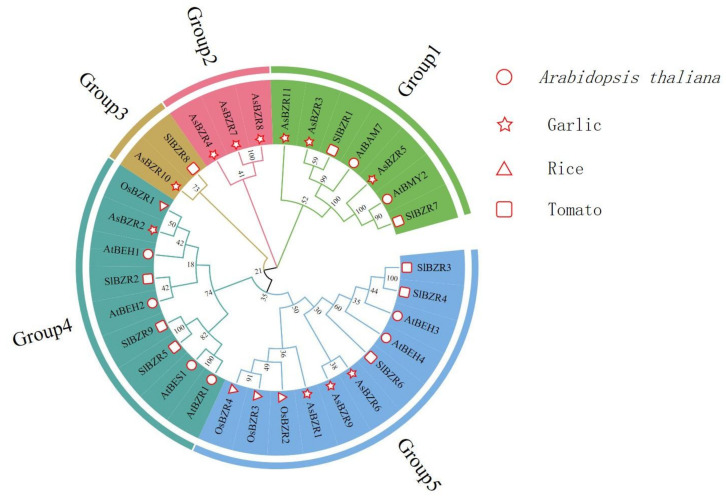
Phylogenetic tree of *BZR* genes from garlic and other species.

**Figure 3 plants-13-02749-f003:**
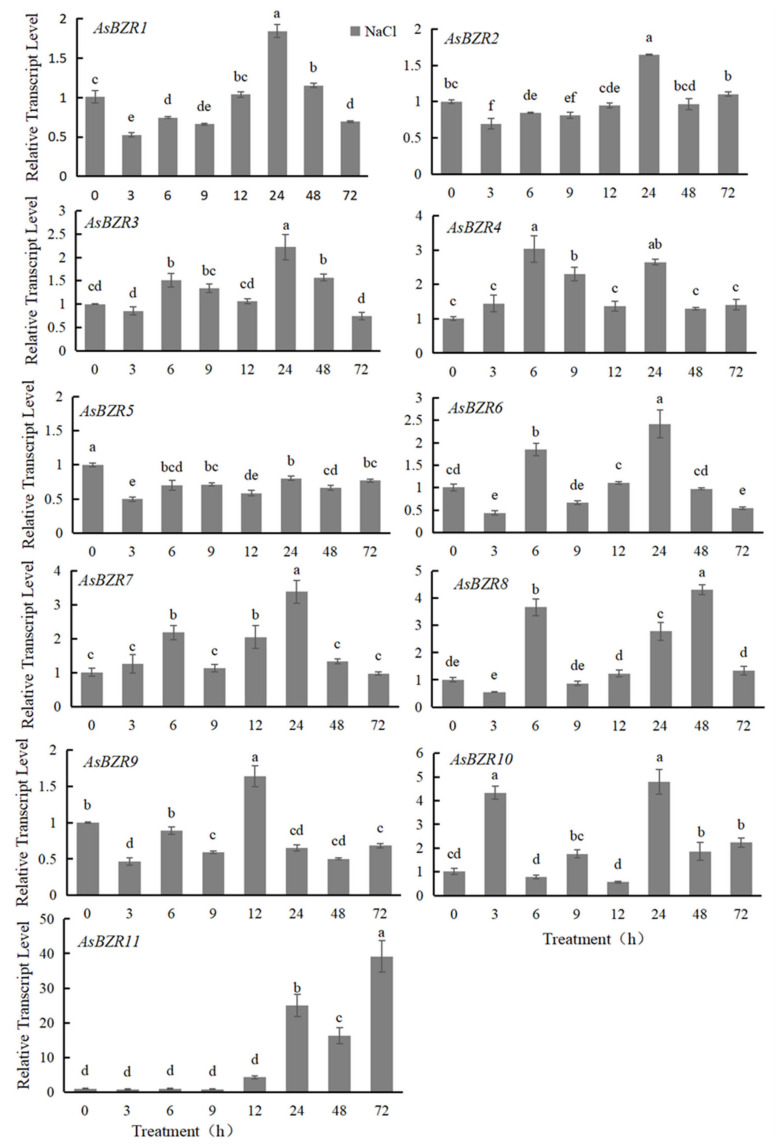
Expression profiles of *AsBZR* genes under salt stress treatment. The numbers below the x-axis represent the time course of the salt stress treatment. Different letters represent significant difference (Tukey HSD, *p* < 0.05).

**Figure 4 plants-13-02749-f004:**
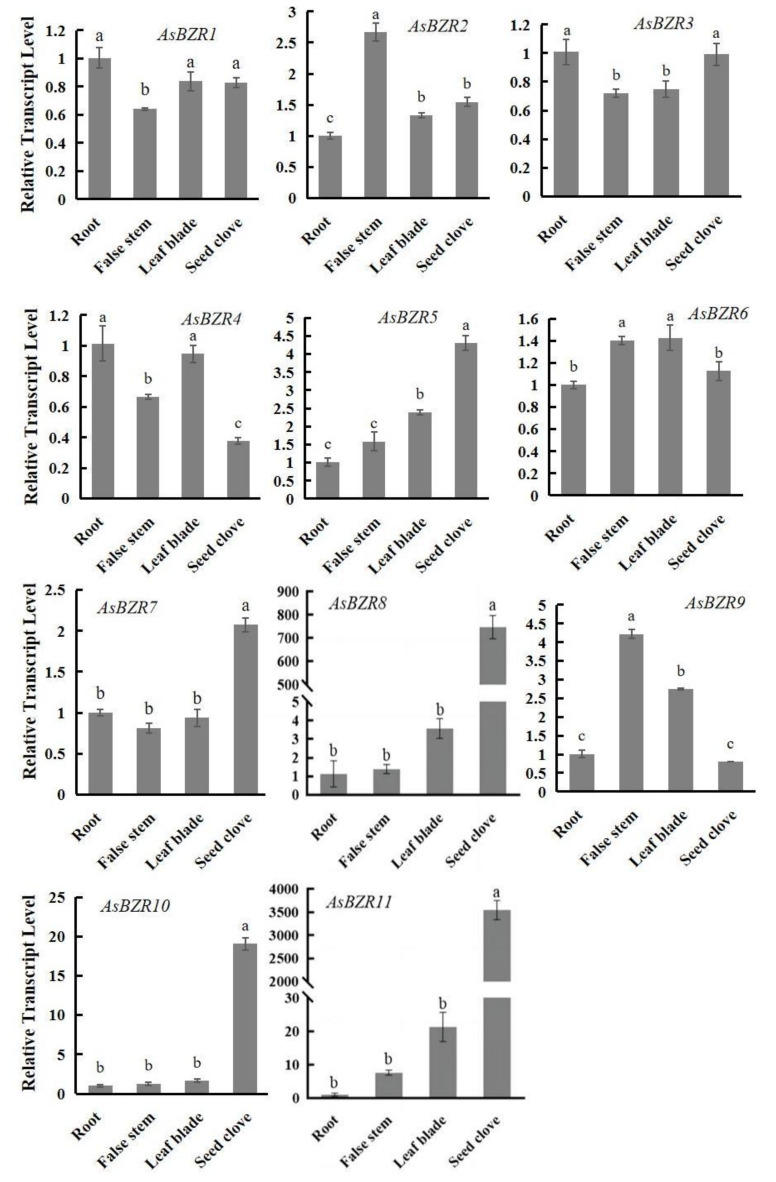
Expression profiles of *AsBZR* genes in different tissues and organs. Different letters represent significant difference (Tukey HSD, *p* < 0.05).

**Figure 5 plants-13-02749-f005:**
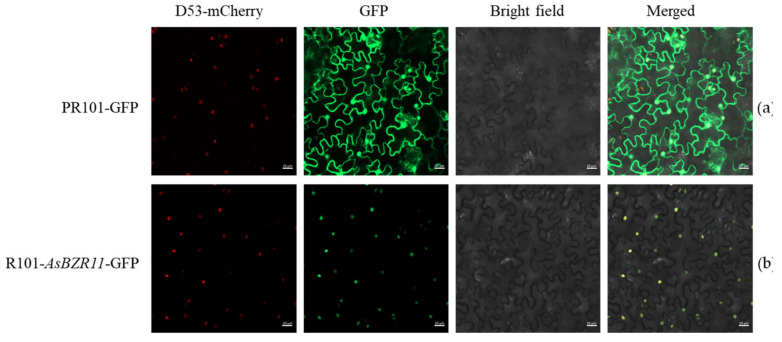
Subcellular localization of the garlic AsBZR11-GFP in *Nicotiana benthamiana* leaves. (**a**) Control vector PR101-GFP; (**b**) *AsBZR11*-PR101-GFP.

**Figure 6 plants-13-02749-f006:**
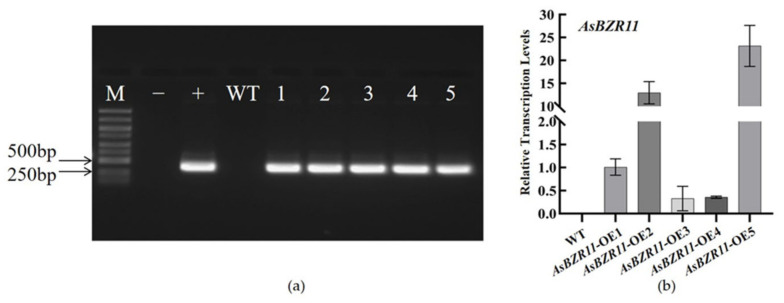
Identification of overexpressed transgenic *Arabidopsis thaliana.* (**a**) *AsBZR11* overexpression in *Arabidopsis* DNA assay; M: DL5000; +: positive control; −: negative control; 1~5 represents five transgenic lines, (**b**) RT–qPCR analysis of *AsBZR11* transcripts in *AsBZR11*-OE lines. Error bars represent standard deviations for the three replicates.

**Figure 7 plants-13-02749-f007:**
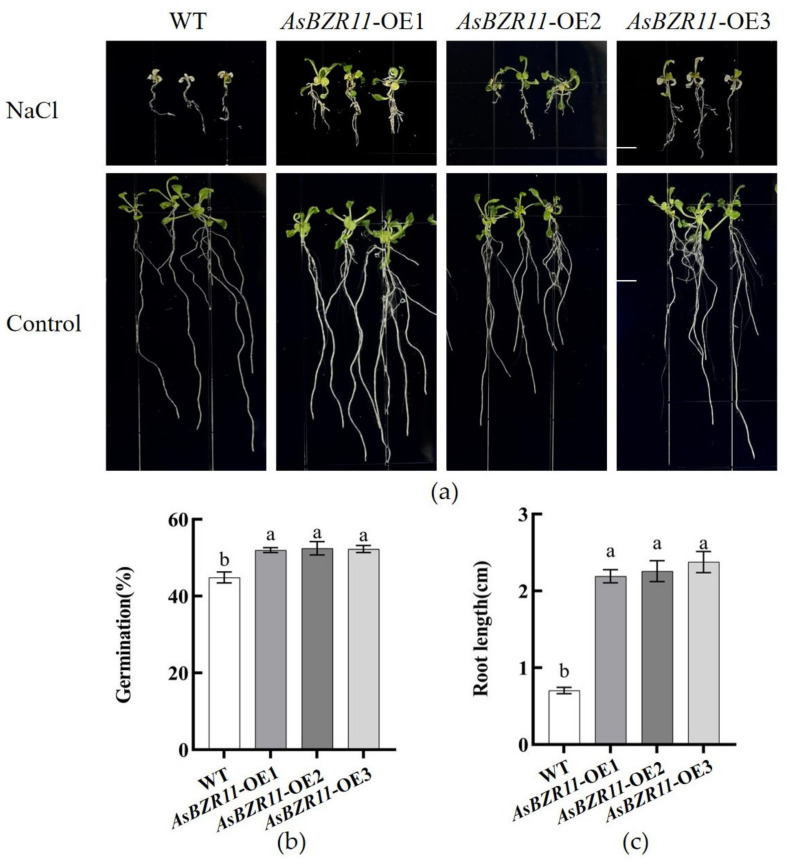
Growth of wild-type and *AsBZR11*-OE lines under salt stress. (**a**) Growth characteristics of wild-type and overexpressing plants under salt stress treatment; (**b**,**c**) Seed germination rates and root length of wild-type and transgenic *Arabidopsis thaliana* treated with 200 mg·L^−1^ NaCl. Different letters represent significant differences at the *p* < 0.05 level. The length of the ruler was 5 mm.

**Figure 8 plants-13-02749-f008:**
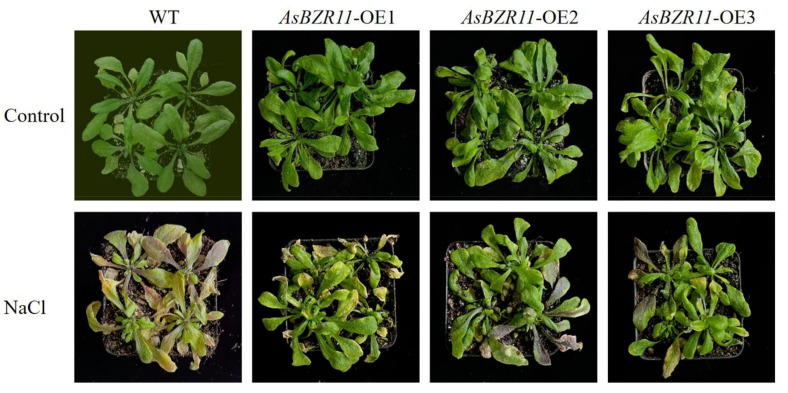
Phenotype of *AsBZR11* overexpression lines under salt stress.

**Figure 9 plants-13-02749-f009:**
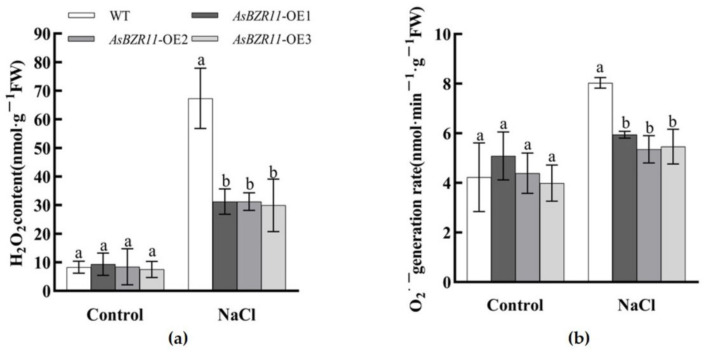
ROS accumulation in *AsBZR11 Arabidopsis* under salt treatment. (**a**) H_2_O_2_ content; (**b**) O_2_^•−^ generation rate. Different letters represent significant difference (Duncan, *p* < 0.05).

**Figure 10 plants-13-02749-f010:**
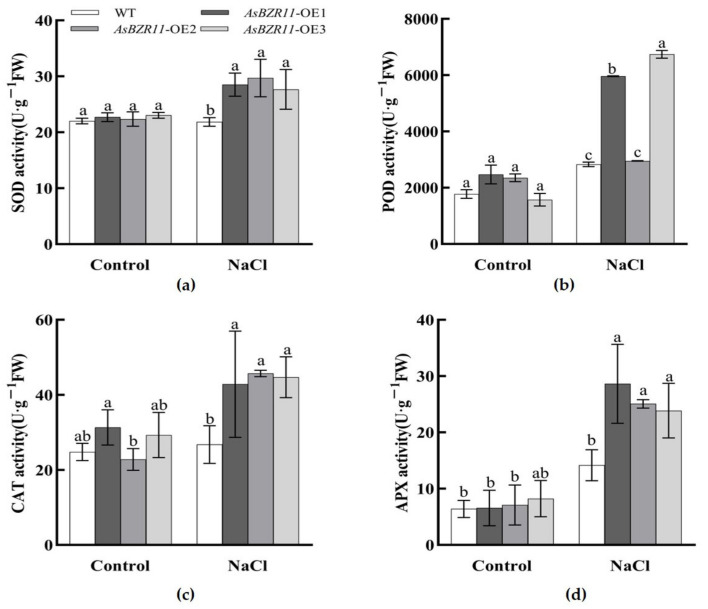
Changes in antioxidant enzyme systems in wild-type and *AsBZR11* transgenic *Arabidopsis* under salt stress. (**a**) SOD activity; (**b**) POD activity; (**c**) CAT activity; (**d**) APX activity. Different letters represent significant difference (Tukey HSD, *p* < 0.05).

**Figure 11 plants-13-02749-f011:**
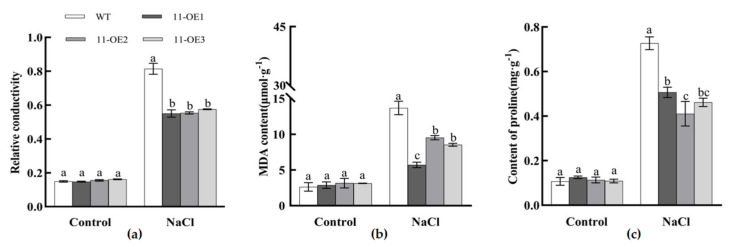
Content of REC, MDA and proline in WT and *AsBZR11*-OE lines under salt stress. (**a**) Relative conductivity; (**b**) MDA content; (**c**) Proline content. Different letters represent significant difference (Tukey HSD, *p* < 0.05).

**Table 1 plants-13-02749-t001:** Primer list for qRT-PCR.

Gene Name	Forward Primer (5′-3′)	Reverse Primer (5′-3′)
*AsBZR1*	GCCTGAGAGCCTACGGGTAACTATAC	ACCCAACCTGCCTCCATACAAAG
*AsBZR2*	CAGGCTGGGATTGTTGAAGAAGATGG	ACTCGGATATGCGGGAGGACATTG
*AsBZR3*	GCGATGGCTATGCTTCAGTTGC	TGGCGACTCTCTTGCTGTTGACTC
*AsBZR4*	TGATCGAAAGCAAGAAAGACACTGGG	TCTCTCATCCGGGGTCATTGAATGG
*AsBZR5*	CACCGTCTCCATCAGTCATCCAG	GCGAGAGCAGCAAGAACATCATTC
*AsBZR6*	TTGTGCGACGAGGCTGGGATG	GACCTTGGACTTGCTGATGTTGATG
*AsBZR7*	TACCTACCACGGGTTTCATTATGCC	CTCATGCTCAATGTCTTCGTCTTCG
*AsBZR8*	ACAAGACCACTCCACTCACCAAC	TGACTCAAGTTAGCGACGACGAC
*AsBZR9*	CGACGAGGCTGGTTGGAGTG	CGACGAGGCTGGTTGGAGTG
*AsBZR10*	GAGAGAGCCGTAGAAGGAGAATCAC	CCATCCAGCCTCTTCGCACAG
*AsBZR11*	CCATCCAGCCTCTTCGCACAG	GTTTAGCCTGAGGTAAGTGAAAGCG
*AsACTIN*	TCCTAACCGAGCGAGGCTACAT	GGAAAAGCACTTCTGGGGCACC

**Table 2 plants-13-02749-t002:** Primer list for subcellular localization.

Primer Name	Primer Sequence (5′-3′)
*AsBZR11*-PR101-GFP-F	tcttcactgttgatacatatgGTTTTGAATACTGCATGGGGATGTAG
*AsBZR11*-PR101-GFP-R	gcccttgctcaccatggatccCATGCATTTTTTGACAAATCGC
PR101-GFP-F	GACGCACAATCCCACTATCC
PR101-GFP-R	CGTCGCCGTCCAGCTCGACCAG

**Table 3 plants-13-02749-t003:** Primer list for vector construction.

Primer Name	Primer Sequence
*AsBZR11*-pBI121-F	gagaacacgggggactctagaGTTTTGAATACTGCATGGGGATGTAG
*AsBZR11*-pBI121-R	ataagggactgaccacccgggCATGCATTTTTTGACAAATCGC

## Data Availability

The original contributions presented in the study are included in the article, further inquiries can be directed to the corresponding author.

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
