# Peer review of "Identification of the BZR Family in Garlic (Allium sativum L.) and Verification of the AsBZR11 under Salt Stress"

_plants, 2024, doi:10.3390/plants13192749_

Round 1

Reviewer 1 Report

Comments and Suggestions for Authors

Summary

The authors have provided a thorough description of the results obtained. However, the manuscript would benefit from additional details regarding the methods employed. Additionally, the Discussion section contains too many introductory concepts and lacks a thorough analysis of the results achieved. English language editing is strongly recommended to enhance the flow and readability of the manuscript.

Abstract

I think it would be more appropriate to refer to the entire class of compounds as "Brassinosteroids" in the plural form.

Lines 11-15: I strongly recommend condensing this paragraph into two lines to enhance conciseness. I suggest focusing more on the setup of your trial and the results obtained

Lines 13-14: The term "Transcription factor" is redundant. Consider using a different term to improve the readability of these lines.

Line 17: Replacing "expression characteristic" with "expression patterns" would be more accurate.

Line 21: Starting a sentence with "And" is not ideal. Consider revising this.

Line 22: Spell out "ROS" in full the first time it is mentioned.

Lines 19-23: This paragraph would benefit from being rewritten to improve its flow and readability.

Line 25: Replace "lay" with "laying" for better grammatical accuracy.

Line 26: Replace "provide" with "providing" to improve the sentence structure.

Introduction

In the Introduction section, as well as throughout the entire manuscript, citations should be formatted with a space before the square bracket, in accordance with the MDPI instructions for authors.

While the section is concise and informative, it would benefit from extensive English editing to improve clarity and readability.

Line 55: Are there specific databases available for garlic DNA or RNA sequences? Is there a reference genome? If so, please incorporate a brief, informative paragraph discussing this topic.

Materials and Methods

Section 4.1.

Please, You must specify the access date (Day, Month, Year) for each website.

Section 4.2.

It would be more appropriate to move this sub-section to the beginning of the Materials and Methods (M&M) section. Additionally, the details regarding the plant material are insufficient. Please provide more comprehensive information about the genotype used, including details such as the provider, cultivar, etc. Further informations are required to ensure clarity and reproducibility of the study.

Line 246: Please correct the garlic scientific name.

Subsection 4.3.

Line 256: Please, provide more details about the kit employed.

Please, add additional details related to the real-time PCR setup. Specify the name of the fluorescent dye employed, the cycles ecc…

Table 1. Table description needs more details.

Table 2. Table description needs more details. Specify the use of lower case.

Table 3. Table description needs more details. Specify the use of lower case.

Line 296: Please, I highly recommend to add a new subsection named “Data analysis”.

Results

Results are clearly presented and Figure a very informative.

Discussion

This section contains a significant amount of introductory information about the differential expression of Brassinosteroids. However, it lacks a thorough comparison of your results with those of other studies. Additionally, the authors should provide a detailed explanation of the underlying “reasons” behind the results obtained. Were there any unexpected outcomes/Results? If Yes, please discuss these and their potential implications.

Conclusion (?)

It is not mandatory, but I think it would be better to incorporate it in the manuscript. 

Comments on the Quality of English Language

English language editing is strongly recommended to enhance the flow and readability of the manuscript.

Author Response

Summary

  1. The authors have provided a thorough description of the results obtained. However, the manuscript would benefit from additional details regarding the methods employed. Additionally, the Discussion section contains too many introductory concepts and lacks a thorough analysis of the results achieved. English language editing is strongly recommended to enhance the flow and readability of the manuscript.

Response: Thanks for your comments. We have supplemented the details of the material method, added a summary language to the discussion and cited the views in other articles.

Abstract

  1. I think it would be more appropriate to refer to the entire class of compounds as "Brassinosteroids" in the plural form.

Response: Thanks for your comments. We agree with the reviewers' suggestions and incorporate the recommended changes into the manuscript.

  1. Lines 11-15: I strongly recommend condensing this paragraph into two lines to enhance conciseness. I suggest focusing more on the setup of your trial and the results obtained

Response: Thanks for your comments. Based on your comments, we have made the corrections to our resubmitted manuscript. For detailed revision, please check all the highlighted sections in manuscript.

  1. Lines 13-14: The term "Transcription factor" is redundant. Consider using a different term to improve the readability of these lines.

Response: Thanks for your comments. As suggested by the reviewer, we have corrected our resubmitted manuscript. For detailed revision, please check all the highlighted sections in manuscript.

  1. Line 17: Replacing "expression characteristic" with "expression patterns" would be more accurate.

Response: Thanks for your comments. We agree with the reviewers' suggestions and incorporate the recommended changes into the manuscript.

  1. Line 21: Starting a sentence with "And" is not ideal. Consider revising this.

Response: Thanks for your comments. In response to the comment, we have corrected the manuscript.

  1. Line 22: Spell out "ROS" in full the first time it is mentioned.

Response: Thanks for your comments. We were really sorry for our mistakes. Thank you for your reminder. As suggested by the reviewer, we have corrected the errors. For detailed revision, please check all the highlighted sections in manuscript (Line 21).

  1. Lines 19-23: This paragraph would benefit from being rewritten to improve its flow and readability.

Response: Thanks for the comments. We are sorry that we did not make this clear in manuscript. In our study, the salt tolerance of overexpression lines was increased, and the germination rate and root length were higher than those of wild type. The content of reactive oxygen species (ROS) decreased and the activity of antioxidant enzymes increased in AsBZR11-OE, suggesting that AsBZR11 has the function of improving the salt tolerance of plants.

  1. Line 25: Replace "lay" with "laying" for better grammatical accuracy.

Response: Thanks for the comments. We were really sorry for our careless mistakes. Thank you for your reminder. As suggested by the reviewer, we have corrected the errors.

  1. Line 26: Replace "provide" with "providing" to improve the sentence structure.

Response: Thanks for the comments. We were really sorry for our careless mistakes. Thank you for your reminder. As suggested by the reviewer, we have corrected the errors.

Introduction

  1. In the Introduction section, as well as throughout the entire manuscript, citations should be formatted with a space before the square bracket, in accordance with the MDPI instructions for authors.

While the section is concise and informative, it would benefit from extensive English editing to improve clarity and readability.

Response: Thanks for the comments.

  1. Line 55: Are there specific databases available for garlic DNA or RNA sequences? Is there a reference genome? If so, please incorporate a brief, informative paragraph discussing this topic.

Response: Thanks for the comments. Sun reported a chromosome-level genome assembly for garlic in 2020, with a total size of approximately 16.24 Gb, as well as the annotation of 57561 predicted protein-coding genes, making garlic the first Allium species with a sequenced genome. We are sorry that we did not make this clear in manuscript. We have added in the manuscript.

Materials and Methods

  1. Section 4.1.

Please, you must specify the access date (Day, Month, Year) for each website.

Response: Thanks for the comments. The HMM profile of the RBZ conserved domain (PF05687) was downloaded from the PFAM database (https://pfam.xfam.org) and used to survey all proteins in June 25, 2023. ExPasy website (http://web.expasy.org/protparam/) was used to survey protein length, molecular weights and isoelectric points of deduced polypeptides in June 28, 2023. The BZR gene family database was downloaded from the Database of Arabidopsis (http://datf.cbi.pku.edu.cn/), rice (https://www.ricedata.cn/gene/), and tomato (https://solgenomics.net/) in July 11, 2023.

  1. Section 4.2.

It would be more appropriate to move this sub-section to the beginning of the Materials and Methods (M&M) section. Additionally, the details regarding the plant material are insufficient. Please provide more comprehensive information about the genotype used, including details such as the provider, cultivar, etc. Further informations are required to ensure clarity and reproducibility of the study.

Response: Thanks for the comments. The plant material used was the garlic cultivar 'Jinxiang' provided by the Laboratory of Vegetable Crop Physiology and Ecology, College of Horticulture, Nanjing Agricultural University.

  1. Line 246: Please correct the garlic scientific name.

Response: Thanks for the comments. Based on your comments, we have made the corrections to our resubmitted manuscript.

Subsection 4.3.

  1. Line 256: Please, provide more details about the kit employed.

Please, add additional details related to the real-time PCR setup. Specify the name of the fluorescent dye employed, the cycles ecc…

Response: Thanks for the comments. qRT-PCR was conducted on an Quantstudio 3 real-time quantitative fluorescent PCR apparatus (AppliedBiosystems, Foster City, CA, USA) using the TOROIVD qRT Master Mix kit (Toroivd, Shanghai, China). The procedure was performed as follows: 95°C 60 s, 95°C 10 s, 60°C 30 s, 40 cycles were carried out.

  1. Table 1. Table description needs more details.

Table 2. Table description needs more details. Specify the use of lower case.

Table 3. Table description needs more details. Specify the use of lower case.

Response: Thanks for the comments. The primer sequences for subcellular localization are shown in Table 2. The primer sequences for overexpression vector are shown in Table 3.

  1. Line 296: Please, I highly recommend to add a new subsection named “Data analysis”.

Response: Thanks for the comments. As suggested, we have add a new subsection named “4.7 Data analysis”.

Results

  1. Results are clearly presented and Figure a very informative.

Response: Thanks for the comments.

Discussion

  1. This section contains a significant amount of introductory information about the differential expression of Brassinosteroids. However, it lacks a thorough comparison of your results with those of other studies. Additionally, the authors should provide a detailed explanation of the underlying “reasons” behind the results obtained. Were there any unexpected outcomes/Results? If Yes, please discuss these and their potential implications.

Response: Thanks for the comments. We tried our best to improve the manuscript and made some changes to the manuscript, which were highlighted in the main text. For detailed revision, please check all the highlighted sections in discuss.

Conclusion (?)

  1. It is not mandatory, but I think it would be better to incorporate it in the manuscript.

Response: Thanks for the comments. We did not add conclusion as a separate summary, we summarized in the discussion.

Reviewer 2 Report

Comments and Suggestions for Authors

The article "Identification of the BZR family in Garlic (Allium sativum L.) and Verification of the AsBZR11 under Salt Stress" by Xianghan Peng et al. covers an exciting and essential topic - plant BZR family. Experiments are well planned, correctly conducted, and well described as results and their interpretation. The authors identified 11 members of the AsBZR gene family in garlic. They characterized one candidate gene, AsBZR11, with salt tolerance function. Then, they describe it in more detail. This study contributes to the understanding of the molecular mechanism of salt tolerance of garlic.

Some small remarks:

Line 31: "Allium Sativum" must be "Allium sativum".

Line 41: "Menthapiperita" must be Mentha piperita.

Line 84: It will be better to be a phylogenetic tree of BZR genes from garlic and other species based on...

Figure 3 and Figure 4: The authors should describe what indicates the small letters above the bars. What statistical analyses were applied to the data? What does "Relative Transcript Level" mean?

Lines 104-105: "Among them, AsBZR1, AsBZR3 and AsBZR4 were mainly concentrated in the roots." Precise this sentence!

What is 1 to 5 in the Figure 6a? It must be written!

Line 153: 10 d must be 10 days!

As shown in Figure ..., it appears in the manuscript ten times!

Line 166: It is the first appearing of these acronyms: SOD, CAT, POD, and APX. In brackets, you must write what they mean!

Under every Figure, the authors must indicate which statistic is used! For example, Figure 10, Figure 11...

Line 246: 'Jinxiang' garlic - Is this a garlic variety, or am I mistaken? It must be clear for the readers!

Line 249: "The salt stress treatment was performed at the time of 5". What does it mean?

Line 263: "The primer for subcellular are shown in Table 2." Will be better to be: The primer sequences for subcellular localization are shown in Table 2.

Table 2: Primer sequences (5'-3') will be better.

Line 272: "The primer sequences are shown in Table 3." Which primer sequences?

What do the abbreviations REC and MDA mean? The authors must write in brackets on the first appearance in the text.

Author Response

  1. Line 31: "Allium Sativum" must be "Allium sativum".

Response: Thanks for the comments. We have modified it. For detailed revision, please check all the highlighted section in manuscript.

  1. Line 41: "Menthapiperita" must be Mentha piperita.

Response: Thanks for the comments. We have modified it. Garlic (Allium sativum L.) is annual and biennial herb of the Allium in the Liliaceae a medicinal and edible, which can be both edible and medicinal

  1. Line 84: It will be better to be a phylogenetic tree of BZR genes from garlic and other species based on...

Response: Thanks for the comments.We have modified it based on your comments.

  1. Figure 3 and Figure 4: The authors should describe what indicates the small letters above the bars. What statistical analyses were applied to the data? What does "Relative Transcript Level" mean?

Response: Thanks for the comments. We agree with the reviewers' suggestions and incorporate the recommended changes into the manuscript.

  1. Lines 104-105: "Among them, AsBZR1, AsBZR3 and AsBZR4 were mainly concentrated in the roots." Precise this sentence!

Response: Thanks for the comments. As suggested by the reviewer, we have modified the description of the sentence. Compared with other members, AsBZR1, AsBZR3 and AsBZR4 are highly expressed in roots.

  1. What is 1 to 5 in the Figure 6a? It must be written!

Response: Thanks for the comments. We agree with the reviewers' suggestions and made some changes to the manuscript, 1~5 represents five transgenic lines.

  1. Line 153: 10 d must be 10 days!

Response: Thanks for the comments. We were really sorry for our careless mistakes. Thank you for your reminder. As suggested by the reviewer, we have corrected the errors.

  1. As shown in Figure ..., it appears in the manuscript ten times!

Response: Thanks for the comments. We agree with the reviewers' suggestions and incorporate the recommended changes into the manuscript.

  1. Line 166: It is the first appearing of these acronyms: SOD, CAT, POD, and APX. In brackets, you must write what they mean!

Response: Thanks for the comments. We are sorry for our carelessness, we have corrected the mistake. After salt stress, the activity levels of SOD (Superoxide dismutase), CAT (Catalase), POD (Peroxidase), and APX (Ascorbate peroxidase) in transgenic plants were significantly higher than those in WT under salt treatment.

  1. Under every Figure, the authors must indicate which statistic is used! For example, Figure 10, Figure 11...

Response: Thanks for the comments. In response to the comment, we have added statistic in note. The experimental data were statistically and analytically analyzed using Excel 2007 and IBM SPSS Statistics 25.0 software, and the significance of the differences be-tween the different time points was tested by Tukey HSD, p<0.05, and graphs were plotted using GraphPad prism 8.0.

  1. Line 246: 'Jinxiang' garlic - Is this a garlic variety, or am I mistaken? It must be clear for the readers!

Response: Thanks for the comments. 'Jinxiang' is a variety of garlic. We have revised it in the manuscript.

  1. Line 249: "The salt stress treatment was performed at the time of 5". What does it mean?

Response: Thanks for the comments. We treated the seedlings of 5 leaves with salt. We were really sorry for our careless mistakes. Thank you for your reminder.

  1. Line 263: "The primer for subcellular are shown in Table 2." Will be better to be: The primer sequences for subcellular localization are shown in Table 2.

Response: Thanks for the comments. Thank you for your reminder. As suggested by the reviewer, we have corrected the errors. For detailed revision, please check all the highlighted sections in manuscript.

  1. Table 2: Primer sequences (5'-3') will be better.

Response: Thanks for the comments. Thank you for your reminder. As suggested by the reviewer, we have revised it.

  1. Line 272: "The primer sequences are shown in Table 3." Which primer sequences?

Response: Thanks for the comments. The primer sequences for overexpression vector are shown in Table 3.

  1. What do the abbreviations REC and MDA mean? The authors must write in brackets on the first appearance in the text.

Response: Thanks for the comments. We are sorry for our carelessness, we have revised it. The MDA (Malondialdehyde) content and REC (Relative conductivity) of wild-type Arabidopsis were significantly higher than those of the overexpression lines.

Round 2

Reviewer 1 Report

Comments and Suggestions for Authors

Authors addressed to all the comments. I only recommend minor english check (If possible, by a native english checker). Than, manuscript will be ready for the submission. 

Author Response

comments:  Authors addressed to all the comments. I only recommend minor English check (If possible, by a native English checker). Then, manuscript will be ready for the submission.

Response: Thanks for the comments. The language of the MS has been carefully checked throughout the paper by all the authors. The revised sections were highlighted in the text, so that you can see all the changes in details. We appreciated your comments, which help us to improve the quality of the MS. We look forward to hearing good news from you.